# Ab-Initio Potential Energy Surfaces by Pairing GNNs with Neural Wave Functions

**Nicholas Gao & Stephan Günnemann**
Department of Informatics & Munich Data Science Institute
Technical University of Munich, Germany
{gaoni,guennemann}@in.tum.de

## Abstract

Solving the Schrödinger equation is key to many quantum mechanical properties. However, an analytical solution is only tractable for single-electron systems. Recently, neural networks succeeded at modeling wave functions of many-electron systems. Together with the variational Monte-Carlo (VMC) framework, this led to solutions on par with the best known classical methods. Still, these neural methods require tremendous amounts of computational resources as one has to train a separate model for each molecular geometry. In this work, we combine a Graph Neural Network (GNN) with a neural wave function to simultaneously solve the Schrödinger equation for multiple geometries via VMC. This enables us to model continuous subsets of the potential energy surface with a single training pass. Compared to existing state-of-the-art networks, our Potential Energy Surface Network (PESNet) speeds up training for multiple geometries by up to 40 times while matching or surpassing their accuracy. This may open the path to accurate and orders of magnitude cheaper quantum mechanical calculations.

## 1 Introduction

In recent years, machine learning gained importance in computational quantum physics and chemistry to accelerate material discovery by approximating quantum mechanical (QM) calculations (Huang & von Lilienfeld, 2021). In particular, a lot of work has gone into building surrogate models to reproduce QM properties, e.g., energies. These models learn from datasets created using classical techniques such as density functional theory (DFT) (Ramakrishnan et al., 2014; Klicpera et al., 2019) or coupled clusters (CCSD) (Chmiela et al., 2018). While this approach has shown great success in recovering the baseline calculations, it suffers from several disadvantages. Firstly, due to the tremendous success of graph neural networks (GNNs) in this area, the regression target quality became the limiting factor for accuracy (Klicpera et al., 2019; Qiao et al., 2021; Batzner et al., 2021), i.e., the network's prediction is closer to the data label than the data label is to the actual QM

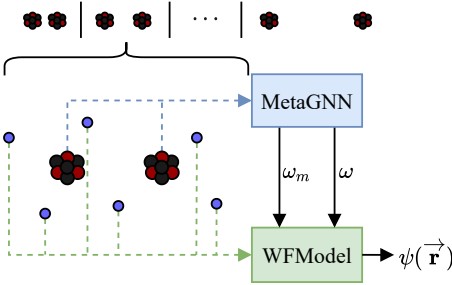

Figure 1: Schematic of PESNet. For each molecular structure (top row), the MetaGNN takes the nuclei graph and parametrizes the WFModel via $\boldsymbol{\omega}$ and $\boldsymbol{\omega}_m$. Given these, the WFModel evaluates the electronic wave function $\psi(\vec{r})$.

property. Secondly, these surrogate models are subject to the usual difficulties of neural networks such as overconfidence outside the training domain (Pappu & Paige, 2020; Guo et al., 2017).

In orthogonal research, neural networks have been used as wave function Ansätze to solve the stationary Schrödinger equation (Kessler et al., 2021; Han et al., 2019). These methods use the variational Monte Carlo (VMC) (McMillan, 1965) framework to iteratively optimize a neural wave function to obtain the ground-state electronic wave function of a given system. Chemists refer to such methods as *ab-initio*, whereas the machine learning community may refer to this as a form of self-generative learning as no dataset is required. The data (electron positions) are sampled from the

wave function itself, and the loss is derived from the Schrödinger equation (Ceperley et al., 1977). This approach has shown great success as multiple authors report results outperforming the traditional 'gold-standard' CCSD on various systems (Pfau et al., 2020; Hermann et al., 2020). However, these techniques require expensive training for each geometry, resulting in high computational requirements and, thus, limiting their application to small sets of configurations.

In this work, we accelerate VMC with neural wave functions by proposing an architecture that solves the Schrödinger equation for multiple systems simultaneously. The core idea is to predict a set of parameters such that a given wave function, e.g., FermiNet (Pfau et al., 2020), solves the Schrödinger equation for a specific geometry. Previously, these parameters were obtained by optimizing a separate wave function for each geometry. We improve this procedure by generating the parameters with a GNN, as illustrated in Figure 1. This enables us to capture continuous subsets of the potential energy surface in one training pass, removing the need for costly retraining. Additionally, we take inspiration from supervised surrogate networks and enforce the invariances of the energy to physical symmetries such as translation, rotation, and reflection (Schütt et al., 2018). While these symmetries hold for observable metrics such as energies, the wave function itself may not have these symmetries. We solve this issue by defining a coordinate system that is equivariant to the symmetries of the energy. In our experiments, our Potential Energy Surface Network (PESNet) consistently matches or surpasses the results of the previous best neural wave functions while training less than $\frac{1}{40}$ of the time for high-resolution potential energy surface scans.

## 2  RELATED WORK

**Molecular property prediction** has seen a surge in publications in recent years with the goal of predicting QM properties such as the energy of a system. Classically, features were constructed by hand and fed into a machine learning model to predict target properties (Christensen et al., 2020; Behler, 2011; Bartók et al., 2013). Lately, GNNs have proven to be more accurate and took over the field (Yang et al., 2019; Klicpera et al., 2019; Schütt et al., 2018). As GNNs approach the accuracy limit, recent work focuses on improving generalization by integrating calculations from computational chemistry. For instance, QDF (Tsubaki & Mizoguchi, 2020) and EANN (Zhang et al., 2019) approximate the electron density while OrbNet (Qiao et al., 2020) and UNiTE (Qiao et al., 2021) include features taken from QM calculations. Another promising direction is $\Delta$-ML models, which only predict the delta between a high-accuracy QM calculation and a faster low-accuracy one (Wengert et al., 2021). Despite their success, surrogate models lack reliability. Even if uncertainty estimates are available (Lamb & Paige, 2020; Hirschfeld et al., 2020), generalization outside of the training regime is unpredictable (Guo et al., 2017).

While such supervised models are architecturally related, they pursue a fundamentally different objective than PESNet. Where surrogate models approximate QM calculations from data, this work focuses on performing the exact QM calculations from first principles.

**Neural wave function Ansätze** in combination with the VMC framework have recently been proposed as an alternative (Carleo & Troyer, 2017) to classical self-consistent field (SCF) methods such as Hartree-Fock, DFT, or CCSD to solve the Schrödinger equation (Szabo & Ostlund, 2012). However, early works were limited to small systems and low accuracy (Kessler et al., 2021; Han et al., 2019; Choo et al., 2020). Recently, FermiNet (Pfau et al., 2020) and PauliNet(Hermann et al., 2020) presented more scalable approaches and accuracy on par with the best traditional QM computations. To further improve accuracy, Wilson et al. (2021) coupled FermiNet with diffusion Monte-Carlo (DMC). But, all these methods need to be trained for each configuration individually. To address this issue, weight-sharing has been proposed to reduce the time per training, but this was initially limited to non-fermionic systems (Yang et al., 2020). In a concurrent work, Scherbela et al. (2021) extend this idea to electronic wave functions. However, their DeepErwin model still requires separate models for each geometry, does not account for symmetries and achieves lower accuracy, as we show in Section 4.

## 3  METHOD

To build a model that solves the Schrödinger equation for many geometries simultaneously and accounts for the symmetries of the energy, we use three key ingredients.

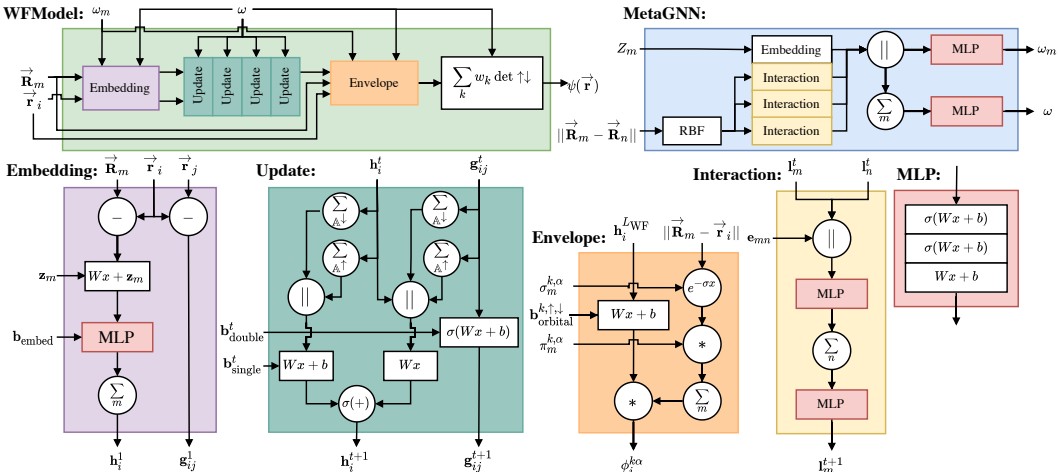

Figure 2: PESNet's architecture is split into two main components, the MetaGNN and the WFModel. Circles indicate parameter-free and rectangles parametrized functions, $\circ\|\circ$ denotes the vector concatenation, $\mathbb{A}^\uparrow$ and $\mathbb{A}^\downarrow$ denote the index sets of the spin-up and spin-down electrons, respectively. To avoid clutter, we left out residual connections.

Firstly, to solve the Schrödinger equation, we leverage the VMC framework, i.e., we iteratively update our wave function model (WFModel) until it converges to the ground-state electronic wave function. The WFModel $\psi_{\boldsymbol{\theta}}(\overrightarrow{\boldsymbol{r}}) : \mathbb{R}^{N \times 3} \mapsto \mathbb{R}$ is a function parametrized by $\boldsymbol{\theta}$ that maps electron configurations to amplitudes. It must obey the Fermi-Dirac statistics, i.e., the sign of the output must flip under the exchange of two electrons of the same spin. As we cover in Section 3.4, the WFModel is essential for sampling electron configurations and computing energies.

Secondly, we extend this to multiple geometries by introducing a GNN that reparametrizes the WFModel. In reference to meta-learning, we call this the MetaGNN. It takes the nuclei coordinates $\overrightarrow{\boldsymbol{R}}_m$ and charges $Z_m$ and outputs subsets $\boldsymbol{\omega}, \boldsymbol{\omega}_m \subset \boldsymbol{\theta}, m \in \{1, \dots, M\}$ of WFModel's parameters. Thanks to message passing, the MetaGNN can capture the full 3D geometry of the nuclei graph.

Lastly, as we prove in Appendix A, to predict energies invariant to rotations and reflections the wave function needs to be equivariant. We accomplish this by constructing an equivariant coordinate system $\boldsymbol{E} = [\overrightarrow{\boldsymbol{e}}_1, \overrightarrow{\boldsymbol{e}}_2, \overrightarrow{\boldsymbol{e}}_3]$ based on the principle component analysis (PCA).

Together, these components form PESNet, whose architecture is shown in Figure 2. Since sampling and energy computations only need the WFModel, a single forward pass of the MetaGNN is sufficient for each geometry during evaluation. Furthermore, its end-to-end differentiability facilitates optimization, see Section 3.4, and we may benefit from better generalization thanks to our equivariant wave function (Elesedy & Zaidi, 2021; Kondor & Trivedi, 2018).

**Notation.** We use bold lower-case letters $\boldsymbol{h}$ for vectors, bold upper-case $\boldsymbol{W}$ letters for matrices, arrows to indicate vectors in 3D, $\overrightarrow{\boldsymbol{r}}_i$ to denote electron coordinates, $\overrightarrow{\boldsymbol{R}}_m, Z_m$ for nuclei coordinates and charge, respectively. $[\circ, \circ]$ and $[\circ]_{i=1}^N$ denote vector concatenations.

## 3.1 Wave Function Model

We use the FermiNet (Pfau et al., 2020) architecture and augment it with a new feature construction that is invariant to reindexing nuclei. In the original FermiNet, the inputs to the first layer are simply concatenations of the electron-nuclei distances. This causes the features to permute if nuclei indexing changes. To circumvent this issue, we propose a new feature construction as follows:

$$\boldsymbol{h}_i^1 = \sum_{m=1}^M \text{MLP}\left(\boldsymbol{W}\left[(\overrightarrow{\boldsymbol{r}}_i - \overrightarrow{\boldsymbol{R}}_m)\boldsymbol{E}, \|\overrightarrow{\boldsymbol{r}}_i - \overrightarrow{\boldsymbol{R}}_m\|\right] + \boldsymbol{z}_m\right), \tag{1}$$

$$\boldsymbol{g}_{ij}^1 = \left((\overrightarrow{\boldsymbol{r}}_i - \overrightarrow{\boldsymbol{r}}_j)\boldsymbol{E}, \|\overrightarrow{\boldsymbol{r}}_i - \overrightarrow{\boldsymbol{r}}_j\|\right) \tag{2}$$

where $\boldsymbol{z}_m$ is an embedding of the $m$-th nuclei and $\boldsymbol{E} \in \mathbb{R}^{3 \times 3}$ is our equivariant coordinate system, see Section 3.3. By summing over all nuclei instead of concatenating we obtain the desired invariance. The features are then iteratively updated using the update rule from Wilson et al. (2021)

$$\boldsymbol{h}_i^{t+1} = \sigma \left( \boldsymbol{W}_{\text{single}}^t \left[ \boldsymbol{h}_i^t, \sum_{j \in \mathbb{A}^\uparrow} \boldsymbol{g}_{ij}^t, \sum_{j \in \mathbb{A}^\downarrow} \boldsymbol{g}_{ij}^t \right] + \boldsymbol{b}_{\text{single}}^t + \boldsymbol{W}_{\text{global}}^t \left[ \sum_{j \in \mathbb{A}^\uparrow} \boldsymbol{h}_j^t, \sum_{j \in \mathbb{A}^\downarrow} \boldsymbol{h}_j^t \right] \right), \quad (3)$$

$$\boldsymbol{g}_{ij}^{t+1} = \sigma \left( \boldsymbol{W}_{\text{double}}^t \boldsymbol{g}_{ij}^t + \boldsymbol{b}_{\text{double}}^t \right) \tag{4}$$

where $\sigma$ is an activation function, $\mathbb{A}^\uparrow$ and $\mathbb{A}^\downarrow$ are the index sets of the spin-up and spin-down electrons, respectively. We also add skip connections where possible. We chose $\sigma := \tanh$ since it must be at least twice differentiable to compute the energy, see Section 3.4. After $L_{\text{WF}}$ many updates, we take the electron embeddings $\boldsymbol{h}_i^{L_{\text{WF}}}$ and construct $K$ orbitals:

$$\phi_{ij}^{k\alpha} = (\boldsymbol{w}_i^{k\alpha} \boldsymbol{h}_j^{L_{\text{WF}}} + b_{\text{orbital},i}^{k\alpha}) \sum_m^M \pi_{im}^{k\alpha} \exp(-\sigma_{im}^{k\alpha} \|\overrightarrow{\boldsymbol{r}}_j - \overrightarrow{\boldsymbol{R}}_m\|), \tag{5}$$

$$\pi_{im}^{k\alpha} = \text{Sigmoid}(p_{im}^{k\alpha}), \qquad \sigma_{im}^{k\alpha} = \text{Softplus}(s_{im}^{k\alpha})$$

where $k \in \{1, \cdots, K\}$, $\alpha \in \{\uparrow, \downarrow\}$, $i, j \in \mathbb{A}^\alpha$, and $\boldsymbol{p}_i, \boldsymbol{s}_i$ are free parameters. Here, we use the sigmoid and softplus functions to ensure that the wave function decays to 0 infinitely far away from any nuclei. To satisfy the antisymmetry to the exchange of same-spin electrons, the output is a weighted sum of determinants (Hutter, 2020)

$$\psi(\overrightarrow{\boldsymbol{r}}) = \sum_{k=1}^K w_k \det \phi^{k\uparrow} \det \phi^{k\downarrow}. \tag{6}$$

## 3.2 METAGNN

The MetaGNN's task is to adapt the WFModel to the geometry at hand. It does so by substituting subsets, $\boldsymbol{\omega}$ and $\boldsymbol{\omega}_m$, of WFModel's parameters. While $\boldsymbol{\omega}_m$ contains parameters specific to nuclei $m$, $\boldsymbol{\omega}$ is a set of nuclei-independent parameters such as biases. To capture the geometry of the nuclei, the GNN embeds the nuclei in a vector space and updates the embeddings via learning message passing. Contrary to surrogate GNNs, we also account for the position in our equivariant coordinate system when initializing the node embeddings to avoid identical embeddings in symmetric structures. Hence, our node embeddings are initialized by

$$\boldsymbol{l}_m^1 = \left[ \boldsymbol{G}_{Z_m}, f_{\text{pos}}(\overrightarrow{\boldsymbol{R}}_m' \boldsymbol{E}) \right] \tag{7}$$

where $\boldsymbol{G}$ is a matrix of charge embeddings, $Z_m \in \mathbb{N}_+$ is the charge of nucleus $m$, $f_{\text{pos}} : \mathbb{R}^3 \mapsto \mathbb{R}^{N_{\text{SBF}} \cdot N_{\text{RBF}}}$ is our positional encoding function, and $\overrightarrow{\boldsymbol{R}}_m' \boldsymbol{E}$ is the relative position of the $m$th nucleus in our equivariant coordinate system $\boldsymbol{E}$ (see Section 3.3). As positional encoding function, we use the spherical Fourier-Bessel basis functions $\tilde{a}_{\text{SBF},ln}$ from Klicpera et al. (2019)

$$f_{\text{pos}}(\overrightarrow{\boldsymbol{x}}) = \sum_{i=1}^3 \left[ \tilde{a}_{\text{SBF},ln}(\|\overrightarrow{\boldsymbol{x}}\|, \angle(\overrightarrow{\boldsymbol{x}}, \overrightarrow{\boldsymbol{e}}_i)) \right]_{l \in \{0,\ldots,N_{\text{SBF}}-1\}, n \in \{1,\ldots,N_{\text{RBF}}\}} \tag{8}$$

with $\overrightarrow{\boldsymbol{e}}_i$ being the $i$th axis of our equivariant coordinate system $\boldsymbol{E}$. Unlike Klicpera et al. (2019), we are working on the fully connected graph and, thus, neither include a cutoff nor the envelope function that decays to 0 at the cutoff.

A message passing layer consists of a message function $f_{\text{msg}}$ and an update function $f_{\text{update}}$. Together, one can compute an update to the embeddings as

$$\boldsymbol{l}_m^{t+1} = f_{\text{update}}^l \left( \boldsymbol{l}_m^t, \sum_n f_{\text{msg}}^t(\boldsymbol{l}_m^t, \boldsymbol{l}_n^t, \boldsymbol{e}_{mn}) \right) \tag{9}$$

where $\boldsymbol{e}_{mn}$ is an embedding of the edge between nucleus $m$ and nucleus $n$. We use Bessel radial basis functions to encode the distances between nuclei (Klicpera et al., 2019). Both $f_{\text{msg}}$ and $f_{\text{update}}$ are realized by simple feed-forward neural networks with residual connections.

After $L_{\text{GNN}}$ many message passing steps, we compute WFModel's parameters on two levels. On the global level, $f_{\text{global}}^{\text{out}}$ outputs the biases of the network and, on the node level, $f_{\text{node}}^{\text{out}}$ outputs nuclei specific parameters:

$$\boldsymbol{\omega} = \left[\boldsymbol{b}_{\text{single/double}}^1, \ldots, \boldsymbol{b}_1^{\uparrow/\downarrow}, \ldots, \boldsymbol{w}\right] \coloneqq f_{\text{global}}^{\text{out}}\left(\left[\sum_m \boldsymbol{l}_m^t\right]_{t=1}^{L_{\text{GNN}}}\right),$$

$$\boldsymbol{\omega}_m = \left[\boldsymbol{z}_m, \boldsymbol{s}_m^{1,\uparrow/\downarrow}, \ldots, \boldsymbol{p}_m^{1,\uparrow/\downarrow}, \ldots\right] \coloneqq f_{\text{node}}^{\text{out}}\left(\left[\boldsymbol{l}_m^t\right]_{t=1}^{L_{\text{GNN}}}\right).$$

(10)

We use distinct feed-forward neural networks with multiple heads for the specific types of parameters estimated to implement $f_{\text{node}}^{\text{out}}$ and $f_{\text{global}}^{\text{out}}$.

### 3.3 EQUIVARIANT COORDINATE SYSTEMS

Incorporating symmetries helps to reduce the training space significantly. In GNNs this is done by only operating on inter-nuclei distances without a clear directionality in space, i.e., without $x, y, z$ coordinates. While this works for predicting observable metrics such as energies, it does not work for wave functions. For instance, any such GNN could only describe spherically symmetric wave functions for the hydrogen atom despite all excited states (the real spherical harmonics) not having such symmetries. Unfortunately, as we show in Appendix B, recently proposed equivariant GNNs (Thomas et al., 2018; Batzner et al., 2021) also suffer from the same limitation.

To solve this issue, we introduce directionality in the form of a coordinate system that is equivariant to rotations and reflections. The axes of our coordinate system $\boldsymbol{E} = \left[\overrightarrow{\boldsymbol{e}}_1, \overrightarrow{\boldsymbol{e}}_2, \overrightarrow{\boldsymbol{e}}_3\right]$ are defined by the principal components of the nuclei coordinates, $\overrightarrow{\boldsymbol{e}}_1^{\text{PCA}}, \overrightarrow{\boldsymbol{e}}_2^{\text{PCA}}, \overrightarrow{\boldsymbol{e}}_3^{\text{PCA}}$. Using PCA is robust to reindexing nuclei and ensures that the axes rotate with the system and form an orthonormal basis. However, as PCA only returns directions up to a sign, we have to resolve the sign ambiguity. We do this by computing an equivariant vector $\overrightarrow{\boldsymbol{v}}$, i.e, a vector that rotates and reflects with the system, and defining the direction of the axes as

$$\overrightarrow{\boldsymbol{e}}_i = \begin{cases} \overrightarrow{\boldsymbol{e}}_i^{\text{PCA}} & \text{, if } \overrightarrow{\boldsymbol{v}}^T \overrightarrow{\boldsymbol{e}}_i^{\text{PCA}} \geq 0, \\ -\overrightarrow{\boldsymbol{e}}_i^{\text{PCA}} & \text{, else.} \end{cases}$$

(11)

As equivariant vector we use the difference between a weighted and the regular center of mass

$$\overrightarrow{\boldsymbol{v}} \coloneqq \frac{1}{M}\sum_{m=1}^{M}\left(\sum_{n=1}^{M}\|\overrightarrow{\boldsymbol{R}}_m - \overrightarrow{\boldsymbol{R}}_n\|^2\right) Z_m \overrightarrow{\boldsymbol{R}}_m',$$

(12)

$$\overrightarrow{\boldsymbol{R}}_m' = \overrightarrow{\boldsymbol{R}}_m - \frac{1}{M}\sum_{m=1}^{M}\overrightarrow{\boldsymbol{R}}_m.$$

(13)

With this construction, we obtain an equivariant coordinate system that defines directionality in space. However, we are aware that PCA may not be robust, e.g., if eigenvalues of the covariance matrix are identical. A detailed discussion on such edge cases can be found in Appendix C.

### 3.4 OPTIMIZATION

We use the standard VMC optimization procedure (Ceperley et al., 1977) where we seek to minimize the expected energy of a wave function $\psi_{\boldsymbol{\theta}}$ parametrized by $\boldsymbol{\theta}$:

$$\mathcal{L} = \frac{\langle\psi_{\boldsymbol{\theta}}|\boldsymbol{H}|\psi_{\boldsymbol{\theta}}\rangle}{\langle\psi_{\boldsymbol{\theta}}|\psi_{\boldsymbol{\theta}}\rangle}$$

(14)

where $\boldsymbol{H}$ is the Hamiltonian of the Schrödinger equation

$$\boldsymbol{H} = -\frac{1}{2}\sum_{i=1}^{N}\nabla_i^2 + \underbrace{\sum_{i=1}^{N}\sum_{j=i}^{N}\frac{1}{\|\overrightarrow{\boldsymbol{r}}_i - \overrightarrow{\boldsymbol{r}}_j\|} - \sum_{i=1}^{N}\sum_{m=1}^{M}\frac{1}{\|\overrightarrow{\boldsymbol{r}}_i - \overrightarrow{\boldsymbol{R}}_m\|} + \sum_{m=1}^{M}\sum_{n=m}^{M}\frac{Z_m Z_n}{\|\overrightarrow{\boldsymbol{R}}_m - \overrightarrow{\boldsymbol{R}}_n\|}}_{V(\overrightarrow{\boldsymbol{r}})}$$

(15)

with $\nabla^2$ being the Laplacian operator and $V(\overrightarrow{r})$ describing the potential energy. Given samples from the probability distribution $\sim \psi_{\boldsymbol{\theta}}^2(\overrightarrow{r})$, one can obtain an unbiased estimate of the gradient

$$
\begin{aligned}
E_{\boldsymbol{\theta}}(\overrightarrow{r}) &= \psi_{\boldsymbol{\theta}}^{-1}(\overrightarrow{r})\boldsymbol{H}\psi_{\boldsymbol{\theta}}(\overrightarrow{r}) \\
&= -\frac{1}{2}\sum_{i=1}^{N}\sum_{k=1}^{3}\left[\frac{\partial^2 \log|\psi_{\boldsymbol{\theta}}(\overrightarrow{r})|}{\partial \overrightarrow{r}_{ik}^2} + \frac{\partial \log|\psi_{\boldsymbol{\theta}}(\overrightarrow{r})|}{\partial \overrightarrow{r}_{ik}}^2\right] + V(\overrightarrow{r}),
\end{aligned}
\tag{16}
$$

$$
\nabla_{\boldsymbol{\theta}}\mathcal{L} = \mathbb{E}_{\overrightarrow{r}\sim\psi_{\boldsymbol{\theta}}^2}\left[\left(E_{\boldsymbol{\theta}}(\overrightarrow{r}) - \mathbb{E}_{\overrightarrow{r}\sim\psi_{\boldsymbol{\theta}}^2}[E_{\boldsymbol{\theta}}(\overrightarrow{r})]\right)\nabla_{\boldsymbol{\theta}}\log|\psi_{\boldsymbol{\theta}}(\overrightarrow{r})|\right]
\tag{17}
$$

where $E_{\boldsymbol{\theta}}(\overrightarrow{r})$ denotes the local energy of the WFModel with parameters $\boldsymbol{\theta}$ for the electron configuration $\overrightarrow{r}$. One can see that for the energy computation, we only need the derivative of the wave function w.r.t. the electron coordinates. As these are no inputs to the MetaGNN, we do not have to differentiate through the MetaGNN to obtain the local energies. We clip the local energy as in Pfau et al. (2020) and obtain samples from $\sim \psi_{\boldsymbol{\theta}}^2(\overrightarrow{r})$ via Metropolis-Hastings. The gradients for the MetaGNN are computed jointly with those of the WFModel by altering Equation 17:

$$
\nabla_{\Theta}\mathcal{L} = \mathbb{E}_{\overrightarrow{r}\sim\psi_{\boldsymbol{\theta}}^2}\left[\left(E_{\boldsymbol{\theta}}(\overrightarrow{r}) - \mathbb{E}_{\overrightarrow{r}\sim\psi_{\boldsymbol{\theta}}^2}[E_{\boldsymbol{\theta}}(\overrightarrow{r})]\right)\nabla_{\Theta}\log|\psi_{\Theta}(\overrightarrow{r})|\right]
\tag{18}
$$

where $\Theta$ is the joint set of WFModel and MetaGNN parameters. To obtain the gradient for multiple geometries, we compute the gradient as in Equation 18 multiple times and average. This joint gradient of the WFModel and the MetaGNN enables us to use a single training pass to simultaneously solve multiple Schrödinger equations.

While Equation 18 provides us with a raw estimate of the gradient, different techniques have been used to construct proper updates to the parameters (Hermann et al., 2020; Pfau et al., 2020). Here, we use natural gradient descent to enable the use of larger learning rates. So, instead of doing a regular gradient descent step in the form of $\Theta^{t+1} = \Theta^t - \eta\nabla_{\Theta}\mathcal{L}$, where $\eta$ is the learning rate, we add the inverse of the Fisher information matrix as a preconditioner

$$
\Theta^{t+1} = \Theta^t - \eta\boldsymbol{F}^{-1}\nabla_{\Theta}\mathcal{L},
\tag{19}
$$

$$
\boldsymbol{F} = \mathbb{E}_{\overrightarrow{r}\sim\psi_{\boldsymbol{\theta}}^2}\left[\nabla_{\Theta}\log|\psi_{\Theta}(\overrightarrow{r})|\nabla_{\Theta}\log|\psi_{\Theta}(\overrightarrow{r})|^T\right].
\tag{20}
$$

Since the Fisher $\boldsymbol{F}$ scales quadratically with the numbers of parameters, we approximate $\boldsymbol{F}^{-1}\nabla_{\Theta}\mathcal{L}$ via the conjugate gradient (CG) method (Neuscamman et al., 2012). To determine the convergence of the CG method, we follow Martens (2010) and stop based on the quadratic error. To avoid tuning the learning rate $\eta$, we clip the norm of the preconditioned gradient $\boldsymbol{F}^{-1}\nabla_{\Theta}\mathcal{L}$ (Pascanu et al., 2013) and use a fixed learning rate for all systems.

We pretrain all networks with the Lamb optimizer (You et al., 2020) on Hartree-Fock orbitals, i.e., we match each of the $K$ orbitals to a Hartree-Fock orbital of a different configuration. During pretraining, only the WFModel and the final biases of the MetaGNN are optimized.

### 3.5 LIMITATIONS

While PESNet is capable of accurately modeling complex potential energy surfaces, we have not focused on architecture search yet. Furthermore, as we discuss in Section 4, PauliNet (Hermann et al., 2020) still offers a better initialization and converges in fewer iterations than our network. Lastly, PESNet is limited to geometries of the same set of nuclei with identical electron spin configurations, i.e., to access properties like the electron affinity one still needs to train two models.

## 4 EXPERIMENTS

To investigate PESNet's accuracy and training time benefit, we compare it to FermiNet (Pfau et al., 2020; Spencer et al., 2020), PauliNet (Hermann et al., 2020), and DeepErwin (Scherbela et al., 2021) on diverse systems ranging from 3 to 28 electrons. Note, the concurrently developed DeepErwin was only recently released as a pre-print and still requires separate models and training for each configuration. While viewing the results on energies one should be aware that, except for PESNet, all methods must be trained separately for each configuration resulting in significantly higher training times, as discussed in Section 4.1.

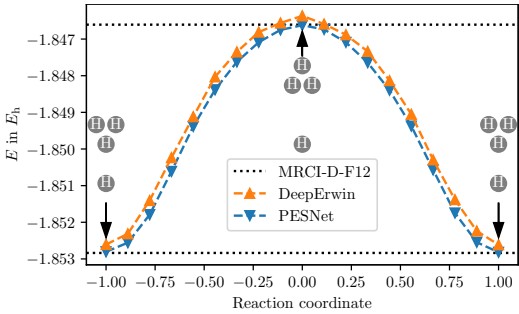
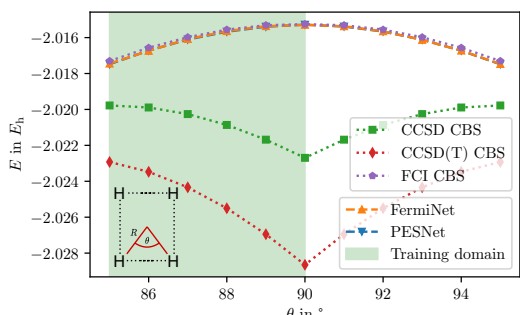

Figure 3: The energy of $H_4^+$ along the first reaction path (Alijah & Varandas, 2008). While PESNet and DeepErwin match the barrier height estimate of the MRCI-D-F12 calculation, PESNet estimates $\approx 0.27\,\mathrm{m}E_\mathrm{h}$ lower energies. Reference data is taken from Scherbela et al. (2021).

Figure 4: Potential energy surface scan of the hydrogen rectangle. Similar to FermiNet, PESNet does not produce the fake minimum at $90°$. Since PESNet respects the symmetries of the energy, we only trained on half of the config space. Reference data is taken from Pfau et al. (2020).

Evaluation of *ab-initio* methods is still a challenge as true energies are rarely known, and experimental data are subject to uncertainties. In addition, many energy differences may seem small due to their large absolute values, but chemists set the threshold for chemical accuracy to $1\,\mathrm{kcal}\,\mathrm{mol}^{-1} \approx 1.6\,\mathrm{m}E_\mathrm{h}$. Thus, seemingly small differences in energy are significant. Therefore, to put all results into perspective, we always include highly accurate classical reference calculations. When comparing VMC methods such as PESNet, FermiNet, PauliNet, and DeepErwin, interpretation is simpler: lower is always better as VMC energies are upper bounds (Szabo & Ostlund, 2012).

To analyze PESNet's ability to capture continuous subsets of the potential energy surface, we train on the continuous energy surface rather than on a discrete set of configurations for potential energy surface scans. The exact procedure and the general experimental setup are described in Appendix D. Additional ablation studies are available in Appendix E.

**Transition path of $H_4^+$ and weight sharing.** Scherbela et al. (2021) use the first transition path of $H_4^+$ (Alijah & Varandas, 2008) to demonstrate the acceleration gained by weight-sharing. But, they found their weight-sharing scheme to be too restrictive and additionally optimized each wave function separately. Unlike DeepErwin, our novel PESNet is flexible enough such that we do not need any extra optimization. In Figure 3, we see the DeepErwin results after their multi-step optimization and the energies of a single PESNet. We notice that while both methods estimate similar transition barriers PESNet results in $\approx 0.27\,\mathrm{m}E_\mathrm{h}$ smaller energies which match the very accurate MRCI-D-F12 results ($\approx 0.015\,\mathrm{m}E_\mathrm{h}$).

**Hydrogen rectangle and symmetries.** The Hydrogen rectangle is a known failure case for CCSD and CCSD(T). While the exact solution, FCI, indicates a local maximum at $\theta = 90°$, both, CCSD and CCSD(T), predict local minima. Figure 4 shows that VMC methods such as FermiNet and our PESNet do not suffer from the same issue. PESNet's energies are identical to FermiNet's ($\approx 0.014\,\mathrm{m}E_\mathrm{h}$) despite training only a single network on half of the configuration space thanks to our equivariant coordinate system.

**The hydrogen chain** is a very common benchmark geometry that allows us to compare our method to a range of classical methods (Motta et al., 2017) as well as to FermiNet, PauliNet, and DeepErwin. Figure 5 shows the potential energy surface of the hydrogen chain computed by a range of methods. While our PESNet generally performs identical to FermiNet, we predict on average $0.31\,\mathrm{m}E_\mathrm{h}$ lower energies. Further, we notice that our results are consistently better than PauliNet and DeepErwin despite only training a single model.

**The nitrogen molecule** poses a challenge as classical methods such as CCSD or CCSD(T) fail to reproduce the experimental results (Lyakh et al., 2012; Le Roy et al., 2006). While the accurate r12-MR-ACPF method more closely matches the experimental results, it scales factorially (Gdanitz, 1998). After Pfau et al. (2020) have shown that FermiNet is capable of modeling such complex triple bonds, we are interested in PESNet's performance. To better represent both methods, we

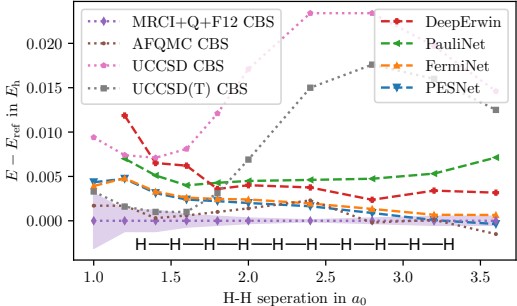
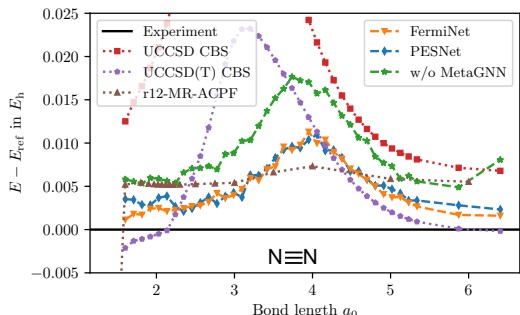

Figure 5: Potential energy surface scan of the hydrogen chain with 10 atoms. We find our PESNet to outperform PauliNet and DeepErwin strictly while matching the results of FermiNet across all configurations. Reference data is taken from Hermann et al. (2020); Pfau et al. (2020); Scherbela et al. (2021); Motta et al. (2017).

Figure 6: Potential energy surface scan of the nitrogen molecule. PESNet yields very similar but slightly higher ($\approx 0.37\,\mathrm{m}E_\mathrm{h}$) energies than FermiNet. Without the MetaGNN the accuracy drops significantly by $\approx 4.3\,\mathrm{m}E_\mathrm{h}$ on average. Reference data is taken from Le Roy et al. (2006); Gdanitz (1998); Pfau et al. (2020).

decided to compare both FermiNet as well as PESNet with 32 determinants due to a substantial performance gain for both methods. The results in Figure 6 show that PESNet agrees very well with FermiNet and is on average just $0.37\,\mathrm{m}E_\mathrm{h}$ higher, despite training only a single model for less than $\frac{1}{47}$ of FermiNet's training time, see Section 4.1. In addition, the ablation of PESNet without the MetaGNN shows a significant loss of accuracy of $4.3\,\mathrm{m}E_\mathrm{h}$ on average.

**Cyclobutadiene and the MetaGNN.** The automerization of cyclobutadiene is challenging due to its multi-reference nature, i.e., single reference methods such as CCSD(T) overestimate the transition barrier (Lyakh et al., 2012). In contrast, PauliNet and FermiNet had success at modelling this challenging system. Naturally, we are interested in how well PESNet can estimate the transition barrier. To be comparable to Spencer et al. (2020), we increased the number of determinants to 32 and the single-stream size to 512. Similar to PauliNet Hermann et al. (2020), we found PESNet to occasionally converge to a higher energy for the transition state depending on the initialization and pretraining. To avoid this, we pick a well-initialized model by training 5 models for 1000 iterations and then continue the rest of the optimization with the model yielding the lowest energy.

As shown in Figure 7, all neural methods converge to the same transition barrier which aligns with the highest MR-CC results at the upper end of the experimental range. But, they require different numbers of training steps and result in different total energies. PauliNet generally converges fastest, but results in the highest energies, whereas FermiNet's transition barrier converges slower but its energies are $70\,\mathrm{m}E_\mathrm{h}$ smaller. Lastly, PESNet's transition barrier converges similar PauliNet's, but its energies are $54\,\mathrm{m}E_\mathrm{h}$ lower than PauliNet's, placing it closer to FermiNet than PauliNet in terms of accuracy. Considering that PESNet has only been trained for $\approx \frac{1}{6}$ of FermiNet's time (see Section 4.1), we are confident that additional optimization would further narrow the gap to FermiNet.

In an additional ablation study, we compare to PESNet without the MetaGNN, i.e, we still train a single model for both states of cyclobutadiene but without weight adaptation. While the results in Figure 7 show that the truncated network's energies continuously decrease, it fails to reproduce the same transition barrier and its energies are $18\,\mathrm{m}E_\mathrm{h}$ worse compared to the full PESNet.

## 4.1 TRAINING TIME

While the previous experiments have shown that our model's accuracy is on par with FermiNet, PESNet's main appeal is its capability to fit multiple geometries simultaneously. Here, we study the training times for all systems from the previous section. We compare the official JAX (Bradbury et al., 2018) implementation of FermiNet (Spencer et al., 2020), the official PyTorch implementation of PauliNet (Hermann et al., 2020), the official TensorFlow implementation of DeepErwin (Scherbela et al., 2021), and our JAX implementation of PESNet. We use the same hyperparameters as in the experiments or the defaults from the respective works. All measurements have been conducted on a machine with 16 AMD EPYC 7543 cores and a single Nvidia A100 GPU. Here, we only mea-

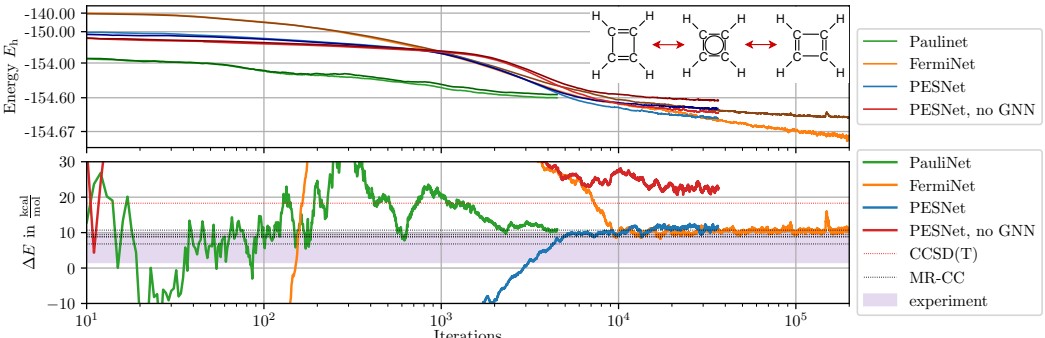

Figure 7: Comparison between the ground and transition states of cyclobutadiene. The top figure shows the total energy plotted in log scale zeroed at $-154.68\,E_h$ with light colors for the ground state and darker colors for the transition state. The bottom figure shows the estimate of the transition barrier. Both figures use a logarithmic x-axis. All neural methods estimate the same transition barriers in line with the highest MR-CC results at the upper end of the experimental data. Reference energies are taken from Hermann et al. (2020); Spencer et al. (2020); Shen & Piecuch (2012).

| | $H_4^+$ (Fig. 3) | H4 (Fig. 4) | H10 (Fig. 5) | N2 (Fig. 6) | Cyclobutadiene (Fig. 7) |
|---|---|---|---|---|---|
| PauliNet | 43h[*] | 34h[*] | 153h | 854h[*] | 437h |
| DeepErwin | 34h | 27h[*] | 111h | — | — |
| FermiNet | 127h[*] | 118h | 594h | 4196h | 2309h |
| PESNet | 20h | 24h | 65h | 89h | 381h |

Table 1: Total GPU (A100) hours to train all models of the respective figures. [*]Experiments are not included in the original works and timings are measured with the default parameters for the respective models. — Larger molecules did not work with DeepErwin.

sure the VMC training time and explicitly exclude the time to perform the SCFs calculations or any pretraining as these take up less than 1% of the total training time for all methods. Furthermore, note that the timings refer to training all models of the respective experiments.

Table 1 shows the GPU hours to train the models of the last section. It is apparent that PESNet used the fewest GPU hours across all systems. Compared to the similar accurate FermiNet, PESNet is up to 47 times faster to train. This speedup is especially noticeable if many configurations shall be evaluated, e.g., 39 nitrogen geometries. Compared to the less accurate PauliNet and DeepErwin, PESNet's speed gain shrinks, but our training times are still consistently lower while achieving significantly better results. Additionally, for H4, H10, and N2, PESNet is not fitted to the plotted discrete set of configurations but instead on a continuous subset of the energy surface. Thus, if one is interested in additional configurations, PESNet's speedup grows linearly in the number of configurations. Still, the numbers in this section do not tell the whole story, thus, we would like to refer the reader to Appendix F and G for additional discussion on training and convergence.

## 5 DISCUSSION

We presented a novel architecture that can simultaneously solve the Schrödinger equation for multiple geometries. Compared to the existing state-of-the-art networks, our PESNet accelerates the training for many configurations by up to 40 times while often achieving better accuracy. The integration of physical symmetries enables us to reduce our training space. Finally, our results show that a single model can capture a continuous subset of the potential energy surface. This acceleration of neural wave functions opens access to accurate quantum mechanical calculations to a broader audience. For instance, it may enable significantly higher-resolution analyses of complex potential energy surfaces with foreseeable applications in generating new datasets with unprecedented accuracy as well as possible integration into molecular dynamics simulations.

**Ethics and reproducibility.** Advanced computational chemistry tools may have a positive impact in chemistry research, for instance in material discovery. However, they also pose the risk of misuse, e.g., for the development of chemical weapons. To the best of our knowledge, our work does not promote misuse any more than general computational chemistry research. To reduce the likelihood of such misuse, we publish our source code under the Hippocratic license (Ehmke, 2019)[1]. To facilitate reproducibility, the source code includes simple scripts to reproduce all experiments from Section 4. Furthermore, we provide a detailed schematic of the computational graph in Figure 2 and additional details on the experimental setup including all hyperparameters in Appendix D.

**Acknowledgements.** We thank David Pfau, James Spencer, Jan Hermann, and Rafael Reisenhofer for providing their results and data, Johannes Margraf, Max Wilson and Christoph Scheurer for helpful discussions, and Johannes Klicpera and Leon Hetzel for their valuable feedback.

Funded by the Federal Ministry of Education and Research (BMBF) and the Free State of Bavaria under the Excellence Strategy of the Federal Government and the Länder.

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

## A INVARIANT ENERGIES REQUIRE EQUIVARIANT WAVE FUNCTIONS

Here, we prove that a wave function needs to be equivariant with respect to rotations and reflections if the energy is invariant. Recall, our goal is to solve the stationary Schrödinger equation

$$\boldsymbol{H}\psi = E\psi \tag{21}$$

where $\psi : \mathbb{R}^{3N} \mapsto \mathbb{R}$ is the electronic wave function. Since the Hamiltonian $\boldsymbol{H}$ encodes the molecular structure via its potential energy term, see Equation 15, a rotation or reflection $\boldsymbol{U} \in O(3)$ of the system results is a unitary transformation applied to the Hamiltonian $\boldsymbol{H} \to \boldsymbol{U}\boldsymbol{H}\boldsymbol{U}^\dagger$. Since the energy $E$ is invariant to rotation and reflection, we obtain the transformed equation

$$\boldsymbol{U}\boldsymbol{H}\boldsymbol{U}^\dagger\psi' = E\psi'. \tag{22}$$

One can see that if $\psi$ is a solution to Equation 21, the equivariantly transformed $\psi \to \boldsymbol{U}\psi$ solves Equation 22

$$\boldsymbol{U}\boldsymbol{H}\boldsymbol{U}^\dagger\boldsymbol{U}\psi = E\boldsymbol{U}\psi, \tag{23}$$
$$\boldsymbol{U}\boldsymbol{H}\psi = E\boldsymbol{U}\psi, \tag{24}$$
$$\boldsymbol{H}\psi = E\psi \tag{25}$$

with the same energy. $\qquad\square$

## B  EQUIVARIANT NEURAL NETWORKS AS WAVE FUNCTIONS

Here, we want to briefly discuss why equivariant neural networks as proposed by Thomas et al. (2018) or Batzner et al. (2021) are no alternative to our equivariant coordinate system. The issue is the same as for regular GNNs (Klicpera et al., 2019), namely that such networks can only represent spherically symmetric functions for atomic systems which, as discussed in Section 3.3, is insufficient for wave functions. While this is obvious for regular GNNs, as they operate only on inter-particle distances rather than vectors, equivariant neural networks take advantage of higher SO(3) representations. However, if one would construct the orbitals $\phi(\overrightarrow{r}) = [\phi_1(\overrightarrow{r}), \dots, \phi_N(\overrightarrow{r})]$ by concatenating $E$ equivariant SO(3) representations $\phi(\overrightarrow{r}) = [\phi_1(\overrightarrow{r}), \dots, \phi_E(\overrightarrow{r})]$ with $\sum_{e=1}^{E} \dim(\phi_e(\overrightarrow{r}_i)) = N$, any resulting real-valued wave function $\psi(\overrightarrow{r}) = \det \phi(\overrightarrow{r})$ would be spherically symmetric, i.e., $\psi(\overrightarrow{r}R) = \psi(\overrightarrow{r}), \forall R \in SO(3)$.

The proof is as follows: If one rotates the electrons $\overrightarrow{r} \in \mathbb{R}^{N \times 3}$ by any rotation matrix $R \in SO(3)$, the orbital matrix changes as

$$\phi(\overrightarrow{r}R) = \phi(\overrightarrow{r})\boldsymbol{D}^R, \tag{26}$$

$$\boldsymbol{D}^R = \operatorname{diag}(\boldsymbol{D}_1^R, \dots, \boldsymbol{D}_E^R) \tag{27}$$

where $\boldsymbol{D}^R \in \mathbb{R}^{N \times N}$ is a block-diagonal matrix and $\boldsymbol{D}_e^R$ is the Wigner-D matrix induced by rotation $R$ corresponding to the $e$-th SO(3) representation. Since Wigner-D matrices are unitary and we restrict our wave function to real-valued

$$\begin{aligned}
\psi(\overrightarrow{r}R) &= \det \phi(\overrightarrow{r}R) \\
&= \det(\phi(\overrightarrow{r})\boldsymbol{D}^R) \\
&= \det \phi(\overrightarrow{r}) \det \boldsymbol{D}^R \\
&= \det \phi(\overrightarrow{r}) \prod_{e=1}^{E} \det \boldsymbol{D}_e^R \\
&= \det \phi(\overrightarrow{r}) \\
&= \psi(\overrightarrow{r}).
\end{aligned} \tag{28}$$

$\square$

## C  EDGE CASES OF THE EQUIVARIANT COORDINATE SYSTEM

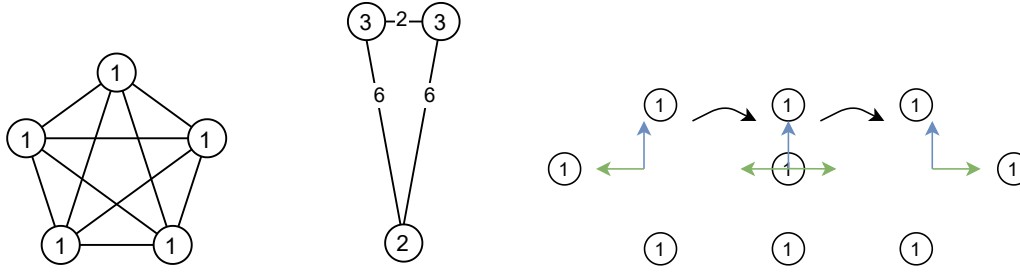

(a) Example of a regular polygon. For any regular polygon on a plane, two eigenvalues of the covariance matrix are going to be identical.

(b) Example of $\overrightarrow{v} = \boldsymbol{0}$.

(c) Example why one can not construct a unique equivariant coordinate system that changes smoothly with changes in the geometry.

Figure 8: Edge cases in the construction of our equivariant coordinate system. Circles indicate nuclei and the numbers their charges.

While the definition of the coordinate system in Section 3.3 works in most cases. There still exist edge cases where the coordinate system may not be unique. To maximize transparency, we discuss some of these cases, when they occur, how we handle them, and what their implications are.

For certain geometries, two eigenvalues of the nuclei covariance matrix might be identical. If that is the case the PCA axes are not unique. This occurs for any geometry where the nuclei coordinates are distributed regularly on a sphere around the center of mass. Examples of such geometries are regular polygons such as the pentagram depicted in Figure 8a. In such cases, we compute the PCA on pseudo coordinates which we obtain by stretching the graph in the direction of the largest Coulomb potential $\frac{Z_m Z_n}{\|\vec{R}_m - \vec{R}_n\|_2}$. In the example from Figure 8a, this is equivalent to stretching the graph along one of the outer edges as they are all of the same lengths. The actual direction does not matter, as it is simply a rotation or reflection of the whole system. While regular spherical patterns are not the only case where this issue arises they are the most common cause.

Another potential issue arises if $\vec{v} = \mathbf{0}$, this occurs for any geometry which is point symmetric in the center of mass such as the pentagram in Figure 8a. In such cases, the signs of the axes do not matter as reflections result in identical geometries. However, there also exist other geometries for which Equation 12 is $\mathbf{0}$. An example is depicted in Figure 8b. But these cases are rare and can be resolved by applying a nonlinear function on the distances in Equation 12.

The occurrence of these edge cases leads us to question: 'Why can we not design a unique coordinate system for each geometry?'. While we ideally would want an equivariant coordinate system that changes smoothly with changes in the geometry, it is impossible. We show a counter example of this in Figure 8c where we see a system of 3 nuclei. In their starting configuration, one can uniquely define two axes as indicated by the colored arrows. But, when moving the leftmost nuclei such that it is in line with the other two nuclei one is left with only one uniquely defined direction as there is no way to differentiate between any orthogonal vector and the blue one. By moving the center nuclei to the right, we again can define two axes for this system, though, one axis is flipped compared to the initial state. So, we neither can define a smoothly changing coordinate system nor a unique one for every system. However, in practice, we do not need a smoothly changing one but only a unique one. While this is already unattainable as shown by the central figure, we also want to stress that any arbitrary orthogonal vector is equivalent due to the symmetries of the system. Considering these aspects, we believe that our coordinate system definition is sufficient in most scenarios.

## D  EXPERIMENTAL SETUP

**Hyperparameters.** If not otherwise specified we used the hyperparameters from Table 2. These result in a similarly sized WFModel to FermiNet from Pfau et al. (2020). For cyclobutadiene, we did not train for the full 60000 iterations but stopped the training after 2 weeks.

**Numerical Stability** To stabilize the optimization, we initialize the last layers of the MetaGNN $f_{node}^{out}$ and $f_{global}^{out}$ such that the biases play the dominant role. Furthermore, we compute the final output of the WFModel in log-domain and use the log-sum-exp trick.

**Learning continuous subsets.** To demonstrate PESNet's ability to capture continuous subsets of the potential energy surface, we train PESNet on a dynamic set of configurations along the energy surface. Specifically, we subdivide the potential energy surface into even-sized bins and place a random walker within each bin. These walkers slightly alter the molecular structure after each step by moving within their bin. This procedure ensures that our model is not restricted to a discrete set of configurations. Therefore, after training, our model can be evaluated at any arbitrary configuration within the training domain without retraining. But, we only evaluate PESNet on configurations where reference calculations are available. This procedure is done for the hydrogen rectangle, the hydrogen chain, and the nitrogen molecule. For $H_4^+$ and cyclobutadiene, we train on discrete sets of geometries from the literature (Scherbela et al., 2021; Kinal & Piecuch, 2007).

**Convergence** is easy to detect in cases where we optimize for a fixed set of configurations as the energy will slowly converge to an optimal value, specifically, the average of the optimal energies. However, in cases where we optimize for a continuous set of configurations, the optimal energy value depends on the current batch of geometries. To still access convergence, we use the fact that the local energy $E_L$ of any eigenfunction (including the ground-state) has 0 variance. So, our convergence criteria is the expected variance of the local energy $\mathbb{E}_{\vec{R} \sim \text{PES}} \left[ \mathbb{E}_{\vec{r} \sim \psi_{\vec{R}}^2} \left[ \left( E_L - \mathbb{E}_{\vec{r} \sim \psi_{\vec{R}}^2} [E_L] \right)^2 \right] \right]$ where the optimal value is 0.

| | Parameter | Value |
|---|---|---|
| Optimization | Local energy clipping | 5.0 |
| Optimization | Batch size | 4096 |
| Optimization | Iterations | 60000 |
| Optimization | #Geometry random walker | 16 |
| Optimization | Learning rate $\eta$ | $\frac{0.1}{(1+t/1000)}$ |
| Natural Gradient | Damping | $10^{-4} \cdot \text{Std}[E_L]$ |
| Natural Gradient | CG max steps | 100 |
| WFModel | Nuclei embedding dim | 64 |
| WFModel | Single-stream width | 256 |
| WFModel | Double-stream width | 32 |
| WFModel | #Update layers | 4 |
| WFModel | #Determinants | 16 |
| MetaGNN | #Message passings | 2 |
| MetaGNN | Embedding dim | 64 |
| MetaGNN | Message dim | 32 |
| MetaGNN | $N_{\text{SBF}}$ | 7 |
| MetaGNN | $N_{\text{RBF}}$ | 6 |
| MetaGNN | MLP depth | 2 |
| MCMC | Proposal step size | 0.02 |
| MCMC | Steps between updates | 40 |
| Pretraining | Iterations | 2000 |
| Pretraining | Learning rate | 0.003 |
| Pretraining | Method | UHF |
| Pretraining | Basis set | STO-6G |
| Evaluation | #Samples | $10^6$ |
| Evaluation | MCMC Steps | 200 |

Table 2: Default hyperparameters.

| | $H_4^+$ | H4 | H10 | N2 | Cyclobutadiene |
|---|---|---|---|---|---|
| no MetaGNN | -1.849286(9) | -2.016199(5) | -5.328944(14) | -109.28322(9) | -154.64419(31) |
| MetaGNN | -1.849363(6) | -2.016208(5) | -5.328916(15) | -109.28570(7) | -154.65469(27) |
| $\Delta E$ | 0.000077(11) | 0.000009(7) | -0.000028(21) | 0.00248(11) | 0.0105(4) |

Table 3: Energies in $E_h$ averaged over the PES for PESNets with and without the MetaGNN. Numbers in brackets indicate the standard error at the last digit(s). In cases without the MetaGNN, we still train a single model for all configurations of a system.

## E ABLATION STUDIES

As hyperparameters often play a significant in the machine learning community, we present some ablation studies in this appendix. All the following experiments only alter one variable at a time, while the rest is fixed as in Table 2. The results in the tables are averaged over the same configurations as in the main body.

Table 3 presents results with and without the MetaGNN. It is noticeable that the gain of the MetaGNN is little to none for small molecules consisting of simple hydrogen atoms. But, for more

| #Dets | $H_4^+$ | H4 | H10 | N2 | Cyclobutadiene |
|---|---|---|---|---|---|
| 16 | -1.849363(6) | -2.016208(5) | -5.328916(15) | -109.28570(7) | -154.65469(27) |
| 32 | -1.849342(4) | -2.016188(6) | -5.328999(13) | -109.28706(7) | -154.65322(27) |
| $\Delta E$ | -0.000021(7) | -0.000019(8) | 0.000083(20) | 0.00136(10) | -0.0015(4) |

Table 4: Energies in $E_h$ averaged over the PES for different number of determinants in our PESNet model. Numbers in brackets indicate the standard error at the last digit(s).

| $\dim(\boldsymbol{h}_i^t)$ | $H_4^+$ | H4 | H10 | N2 | Cyclobutadiene |
|---|---|---|---|---|---|
| 256 | -1.849363(6) | -2.016208(5) | -5.328916(15) | -109.28570(7) | -154.65469(27) |
| 512 | -1.8493543(28) | -2.016190(7) | -5.328794(17) | -109.28662(6) | -154.65042(28) |
| $\Delta E$ | -0.000009(7) | -0.000017(8) | -0.000122(23) | 0.00092(9) | -0.0043(4) |

Table 5: Energies in $E_h$ averaged over the PES for different single-stream sizes in our PESNet model. Numbers in brackets indicate the standard error at the last digit(s).

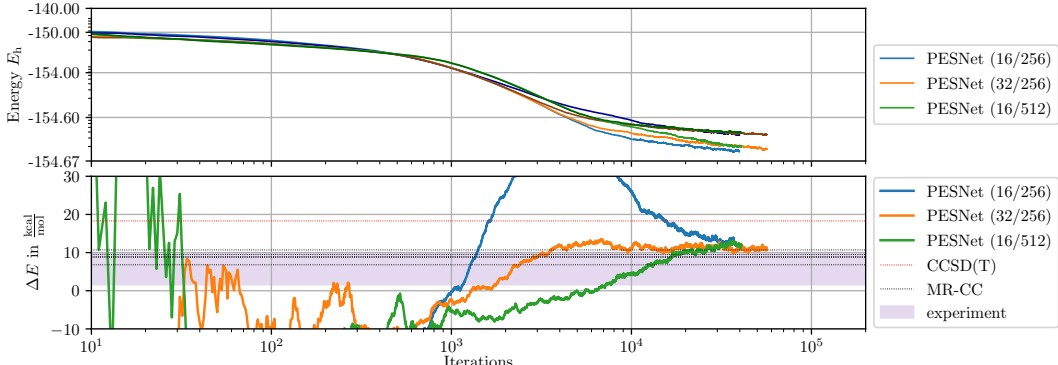

Figure 9: Comparison of different PESNet configurations on cyclobutadiene. The configurations are named (#determinant/single-stream width) with light colors for the ground state and darker colors for the transition state.

complex molecules such as nitrogen and cyclobutadiene, we notice significant improvements of $2.5\,mE_h$ and $10.5\,mE_h$, respectively. Moreover, the MetaGNN enables us to account for symmetries of the energy while the WFModel itself is only invariant w.r.t. to translation but not to rotation, reflection, and reindexing of nuclei.

Table 4 shows the impact of the number of determinants on the average energy for the systems from Section 4. For small hydrogen-based systems, the number of determinants is mostly irrelevant, while larger numbers of determinants improve performance for nitrogen. But, this does not seem to carry over to cyclobutadiene. While the total estimated energy is higher for the larger model, we noticed that it is significantly faster at converging the transition barrier. Figure 9 illustrates this by comparing the convergence of the 16 and 32 determinant models.

Increasing the single-stream size does not seem to result in any benefit for most models as Table 5 shows. Again, the hydrogen systems are mostly unaffected by this hyperparameter while nitrogen benefits from larger hidden dimensions but cyclobutadiene converges worse. We suspect that this is due to the optimization problem becoming significantly harder. Firstly, increasing the WFModel increases the number of parameters the MetaGNN has to predict. Secondly, we estimate the inverse of the Fisher with a finite fixed-sized batch but the Fisher grows quadratically with the number of parameters which also grow quadratically with the single size stream.

## F  TIME PER ITERATION

While the main document already covers the time it took to reproduce the results from the figures, we want to use this appendix to provide more details on this. Table 6 lists the time per iteration for a single model instead of the whole training time to reproduce the potential energy surfaces as in Table 1. While we find these numbers to be misleading we still want to disclose them to support open research. The main issue with these numbers is that they do not take the quality of the update into account, e.g., PauliNet and DeepErwin are trained with Adam (Kingma & Ba, 2014), FermiNet with K-FAC (Martens & Grosse, 2015), and PESNet with CG-computed natural gradient (Neuscamman et al., 2012). This has implications on the number of iterations one has to train. For Table 1 in the main body, we assumed that PauliNet is trained for 10000 iterations (Hermann et al., 2020), Deep-Erwin for 7000 iterations (Scherbela et al., 2021), FermiNet for 200000 iterations (Pfau et al., 2020),

|  | $H_4^+$ | H4 | H10 | N2 | Cyclobutadiene |
|---|---|---|---|---|---|
| PauliNet | 0.83s | 1.13s | 5.51s | 8.09s | 175s |
| DeepErwin | 0.92s | 1.28s | 4.88s | — | — |
| FermiNet | 0.12s | 0.19s | 1.07s | 1.99s | 20.8s |
| PESNet | 1.19s | 1.42s | 3.91s | 5.32s | 33.2s |

Table 6: Time per training step.

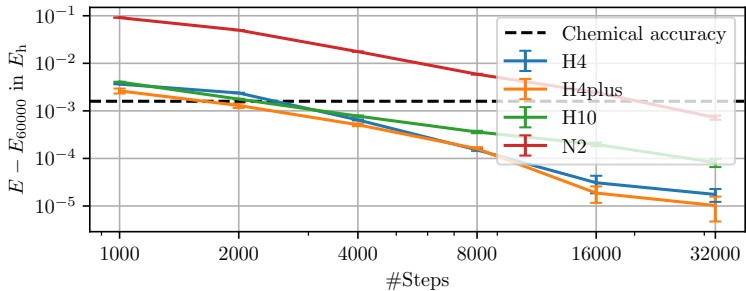

Figure 10: Convergence behavior of PESNet. Error bars indicate the standard error of the mean.

and PESNet for 60000 iterations. Though, one can assume that neither PauliNet nor DeepErwin are going to produce similar results to FermiNet and PESNet on the more complex nitrogen molecule or cyclobutadiene in 10000 or 7000 iterations. This is further discussed in Appendix G. When viewing these results, one also has to keep in mind the different quality of the results, e.g., FermiNet and PESNet strictly outperform PauliNet and DeepErwin on all systems. Another potential issue arises due to the choice of deep learning framework the models are implemented in. It has been shown that JAX works very well for computing the kinetic energy (Spencer et al., 2020) which usually is the largest workload when computing an update.

## G CONVERGENCE

When choosing a method to solve the Schrödinger equation one has to pick between accuracy and speed. For classical methods, this might be the difference between choosing DFT, CCSD(T), or FCI. In the neural VMC setting, one way to reflect this is by choosing the number of training steps. To better investigate this, we present convergence graphs for H4, $H_4^+$, H10, and N2 in Figure 10. For cyclobutadiene, we can see the convergence of different configurations of PESNet in Figure 9. One can see that our network converges quite quickly on hydrogen-based systems such as the hydrogen rectangle, $H_4^+$, or the hydrogen chain. For these small systems, PESNet surpasses PauliNet's and DeepErwin's accuracy in less than 8000 training steps, reducing the training times to 2.9h, 3.2h, and 8.7h, respectively. In less than 16000 steps, about 17.4h of training, PESNet surpasses FermiNet on the hydrogen chain. For more complicated molecules such as N2 and cyclobutadiene, the training requires more iterations to converge. So, we may also expect the extrapolated numbers from Table 1 for PauliNet to be an optimistic estimate.

