# OpenReview forum: "Ab-Initio Potential Energy Surfaces by Pairing GNNs with Neural Wave Functions"
_ICLR.cc/2022/Conference — ICLR 2022 Spotlight_

### Official Review · Reviewer_3XzF · 2021-11-02

**Correctness:** 3
**Technical Novelty And Significance:** 3
**Empirical Novelty And Significance:** 3
**Recommendation:** 8
**Confidence:** 4

**Main Review:**

Overall, the manuscript is well written and theory and experiments are presented in a concise fashion. The idea of coupling neural wavefunctions for different nuclear configurations with a meta-GNN is promising, since this allows for simultaneous solution with VMC. The experiments demonstrate the utility of such an approach in terms of accuracy and computational speed up.  There are, however, some issues that should be addressed.

First, the comparison of computational timings between the different methods is hard to follow at points. Is it correct to assume that the reported timings are those needed to solve for every configuration of a particular molecule? How is this handled in case of the systems where PESNet is solved for a continuous subset of structures (e.g. are there global convergence criteria)? Regarding the time per iteration reported in the SI, it is unclear how exactly these are computed. For example, in Table 1, it is assumed that 10000 iterations with PauliNet for H4+ take 43 hours, which should give a time per iteration of ~15 s instead of the 0.83 s reported. Similarly, solving PESNet for the same system takes only half the time in Table 1, while the SI reports 60000 iterations @ 1.29 s. Finally, there is  the question whether the pretraining on HF orbitals is included in the timings.

Second, the comparison between different approaches could profit from additional information. While not a completely reliable measure, it would be helpful to report the parameter count of the different models as used in the experiments in order to gain an impression of their overall flexibility. Moreover, an ablation study, where the MetaGNN and WFModel are optimized for the different configurations of a system individually would add to the analysis of the model. This would help to identify how the MetaGNN influences the wavefunction solution of a single conformer and if e.g. the structural dependence of the weights introduced in the MetaGNN is beneficial.

Finally, regarding the the equivariant coordinate frame. The PCA based frame can potentially become unstable if structures are strongly distorted (e.g. principal axis change order). Perhaps the authors can comment on how this is handled and whether it presents a problem for the wavefunction model or limits the range of structural configurations that can be studied with a single metaGNN. Is the same coordinate frame used for all configurations of a system or are individual frames employed? Finally, it would be interesting to motivate why this particular approach of introducing SE(3) equivariance was chosen in favor of more conventional SE(3) equivariant graph networks.

**Summary Of The Paper:**

The authors introduce an approach based on using graph neural networks to parametrize wavefunctions represented by an equivariant neural network. This allows them simultaneously solve for the ground state energy of multiple molecular configurations with variational Monte Carlo.

**Summary Of The Review:**

The manuscript is well written and the proposed approach is sufficiently novel. Its advantages are further clearly demonstrated in a series of experiments. As such, I recommend to accept this contribution.

---

> ### Author Response · Authors · 2021-11-12
> **Official Response to Reviewer 3XzF - Part 1**
>
> Thank you for taking the time to review our work. We appreciate your feedback and are glad that you are convinced by the accuracy and speed up of our method.
>
> We are happy to discuss your issues and hope that we can clear up any misunderstandings.
> >__Is it correct to assume that the reported timings are those needed to solve for every configuration of a particular molecule?__
>
> Yes, the timings are measured by the __overall__ training time for each experiment, so `number_of_configs x number_of_iterations x time_per_iteration` for DeepErwin, PauliNet, and FermiNet.
>
> > __For example, in Table 1, it is assumed that 10000 iterations with PauliNet for H4+ take 43 hours, which should give a time per iteration of ~15 s instead of the 0.83 s reported.__
>
> * The results in Appendix E refer to the time per update per model.
> * PauliNet takes much longer in Table 1 because we have to train `number_of_configs` models.
>
>
> > __Finally, there is the question whether the pretraining on HF orbitals is included in the timings.__
>
> Neither the pretraining nor the SCFs calculations are included in any method since they only account for less than 1% of the training time for all methods and differ vastly between methods.
>
> To increase clarity on the timings, we added in Section 4.1:
> >*Here, we only measure the VMC training time and explicitly exclude the time to perform the SCFs calculations or any pretraining as these take up less than 1% of the total training time for all methods. Further, note that the timings refer to training all models of the respective experiments.*
>
> and in Appendix E:
> > Table 6 lists the time per iteration *for a single model* instead of the whole training time *to reproduce the potential energy surfaces* as in Table 1.
>
> ---
>
>
> > __Similarly, solving PESNet for the same system takes only half the time in Table 1, while the SI reports 60000 iterations @ 1.29 s.__
>
> Thank you for catching this. We unfortunately printed outdated numbers for PESNet in Table 6. Please mind that these changes do not affect our conclusions as the numbers in Table 1 are still correct. Here are the changes that have been made to Table 6:
>
> | |before|after|
> |---|---|---|
> |H4plus|1.29s|1.19s|
> |N2|4.74s|5.32s|
> |Cyclobutadiene|18.31s|33.21s|
>
> ---
>
> > __How is this handled in case of the systems where PESNet is solved for a continuous subset of structures (e.g. are there global convergence criteria)?__
>
> * Since PESNet's innovation lies in training a single model for all configurations `number_of_configs=1`.
> * To detect convergence, one can focus on the expected standard deviation of the local energy across the different configurations $\mathbb{E}_\text{configs}[\sigma_\text{electrons}[E_L]]$, because the local energy of any eigenfunction (and, thus, also the ground-state) of the Hamilton has 0 variance. A small paragraph on this has been added to Appendix C.
>
> ---
>
> > __While not a completely reliable measure, it would be helpful to report the parameter count of the different models as used in the experiments in order to gain an impression of their overall flexibility.__
>
> We are happy to share the parameter counts of the respective models:
>
> |WFModel (PESNet)| MetaGNN (PESNet)|FermiNet|PauliNet|DeepErwin|
> |---|---|---|---|---|
> |~700k-800k|~1.1m|~700-800k| ~150k| ~90k|
>
>
> Please note that the runtimes reported in our paper are much more reflective of the model cost than parameter counts are.
>
> ---
>
> > __Moreover, an ablation study, where the MetaGNN and WFModel are optimized for the different configurations of a system individually would add to the analysis of the model.__
>
> While we already performed several ablation studies, among other things PESNet without the MetaGNN, we agree that such an experiment would be interesting. Unfortunately, it is out of reach for most research groups (~5000h A100 GPU hours) including us. Considering that the WFModel does not change in expressivity whether the MetaGNN is present or not, we expect the model to behave very similar to FermiNet due to the architectural similarity of the WFModel.

---

> > ### Comment · Reviewer_3XzF · 2021-11-22
> > **Response**
> >
> > Thank you for the clarifying comments.

---

> ### Author Response · Authors · 2021-11-12
> **Official Response to Reviewer 3XzF - Part 2**
>
>
> > __The PCA based frame can potentially become unstable if structures are strongly distorted (e.g. principal axis change order).__
>
> We are aware of these instabilities of PCA and discuss in Appendix B that we only need a unique coordinate system per configuration and not a smoothly changing one. In fact, there exists no coordinate system that changes smoothly with changes in geometry. If one encounters a system where a unique coordinate system cannot be defined one can perform a perturbation by some very small epsilon (1e-7) such that a unique orientation exists and the energy remains the same.
>
> ---
>
> >__Is the same coordinate frame used for all configurations of a system or are individual frames employed?__
>
> We use individual frames for each configuration to ensure that the coordinate system is equivariant if one inputs two rotations of the same geometry.
>
> ---
>
> > __Finally, it would be interesting to motivate why this particular approach of introducing SE(3) equivariance was chosen in favor of more conventional SE(3) equivariant graph networks.__
>
> Great question! While at first sight SE(3) equivariant neural networks have all the symmetries we are interested in, they too can only model spherically symmetric functions which are insufficient as briefly explained in Section 3.3.
> To give mathematical reasoning, one can show that the use of SO(3) representations to construct the orbital matrices $\phi(r)$ still results in spherically symmetric functions $\psi(r)=\det\phi(r)$. Let us assume that we view a single atom in isolation and $\phi(r)=[\phi_1(r), ..., \phi_n(r)], \phi_i(r)\in\mathbb{R}^n$ is the orbital matrix where $\phi_i(r)$ may be part of an equivariant SO(3) representation. If the electrons are rotated by $R$ around the nuclei, the orbital matrix changes as
> $$
> \phi(Rr) = \phi(r)U, U = [D_1, ..., D_k]
> $$
> where $U$ is a block-diagonal matrix of Wigner-D matrices $D_i$. Thus, all real-valued functions $\psi$ do not change under electron rotations
> $$
> \psi(Rr) = \det \phi(Rr) = \det \phi(r)U = \det\phi(r)\det U = \psi(r) \prod\det D_i = \psi(r)
> $$
> since Wigner-D matrices are unitary. Thus, wave functions of SE(3) equivariant NN are still spherically symmetric.
>
> In the updated paper, we include a brief discussion on why equivariant coordinate frames and not equivariant networks are chosen.

---

### Official Review · Reviewer_bjEg · 2021-11-02

**Correctness:** 3
**Technical Novelty And Significance:** 3
**Empirical Novelty And Significance:** 2
**Recommendation:** 8
**Confidence:** 4

**Main Review:**

**Strengths**:
* Comprehensive set of tests and comparison to other relevant work
* Study of relevant hyperparameters and convergence analysis
* Variational energies are respectable across all studied problems

**Weaknesses**:
* A major improvement of the method (compared to existing approaches like FermiNet) is that the model can be optimized on multiple similar configurations simultaneously to achieve faster net runtime. While it is an interesting observation that GNN is capable of conditioning the model parameters on the geometry to adjust the modeled ground state wave function, other methods could use optimization results from one configuration as a starting point for others. This generally speeds up the number of iterations needed to converge to slightly different solutions. Such an estimate would be a more fair approach to runtime comparison. I believe that even if the runtime advantage is noticeably reduced it is still an interesting result.
* The other stress test to justify that the GNN model has learned something non-trivial would be to include a baseline method where the parameters of the wave function are not changing with the system. (e.g. show the results of PESNet for Fig. 3 and Fig. 4 with single parameters w; w_m predicted by GNN intermediate regime). This would enable a clear assessment of how relevant the modifications to the ground state are.


**Summary Of The Paper:**

The paper develops a neural network based variational ansatz for modeling wave functions. Authors build their model on top of FermiNet architecture with a few modifications: they use a different feature embedding approach that is invariant with respect to basic spatial symmetries. In addition, authors use a GNN “hypernetwork” that predicts the parameters of the variational wave function for a given configuration of the system. With these modifications authors show that the model can be optimized on a range of system configurations simultaneously. The optimization is performed in a standard VMC setting.

The results are compared on commonly studied problems such as variations of energy of the H4 molecule, hydrogen chain system, nitrogen molecule and cyclobutadiene. Authors find that their approach generally achieves similar results to those of a FermiNet model, while training on multiple system configurations simultaneously, reducing the time needed to obtain variational energies for multiple configurations.

**Summary Of The Review:**

The paper presents sufficiently strong results and an interesting demonstration of how a single network can be used to simultaneously approximate a family of ground state wave functions. Results are compared to relevant baselines and with simple additional checks suggested in the main review should be a good contribution to variational methods in quantum chemistry.

**Update**
After the discussion period many of my concerns have been cleared. Changing my score from 6 to 8.

---

> ### Author Response · Authors · 2021-11-12
> **Official Response to Reviewer bjEg**
>
> Thank you for your review, we appreciate your time and are glad that you find our manuscript a good contribution to quantum chemistry.
> To address your concerns:
> > __..., other methods could use optimization results from one configuration as a starting point for others. This generally speeds up the number of iterations needed to converge to slightly different solutions.__
>
> We fully agree with your comment. For example, DeepErwin [1] is using this principle. Note, however, that we include their results in our paper (despite the pre-print status of their work). Realizing this principle for the other methods, however, is out of reach with common compute resources (i.e. it would require ~6000-8000 A100 GPU hours). Moreover, while such a scheme certainly aids when dealing with hydrogen systems we do not expect large benefits for more complicated systems like N2 or cyclobutadiene. Due to the high-frequency core wave function of heavy atoms, the majority of optimization is spent fine-tuning in the range of millihartrees which has to be redone once the geometry changes.
>
>
> > __The other stress test to justify that the GNN model has learned something non-trivial would be to include a baseline method where the parameters of the wave function are not changing with the system.__
>
> We believe that our ablation studies of PESNet without the MetaGNN are the experiments you are describing. In these experiments (Figure 7 and Table 3), we train a single PESNet without the MetaGNN for all configurations which is equivalent to having only one instance of $\omega$ and $\omega_m$ for multiple geometries. Both results suggest that the MetaGNN is essential to ensure SOTA results on challenging systems (with MetaGNN  $\sim 2.5\text{m}E_\text{h}$ and $\sim 18\text{m}E_\text{h}$ better results on N2 and cyclobutadiene, respectively).
> However, we recognize that the labels may not have been concise enough, thus, we added some clarification in Section 4:
> > In an additional ablation study, we compare to PESNet without the MetaGNN. *Please note that we still only train a single model for both states of cyclobutadiene.*
>
> and in the label of Table 3:
> > *In the cases without the MetaGNN, note that we still train a single model for all states of a system.*
>
> ---
> [1] Michael Scherbela, Rafael Reisenhofer, Leon Gerard, Philipp Marquetand, and Philipp Grohs. Solving the electronic Schrödinger equation for multiple nuclear geometries with weight-sharing deep neural networks

---

> > ### Comment · Reviewer_bjEg · 2021-11-29
> > **Comment**
> >
> > Thanks for your reply!
> >
> > What described in DeepErwin is Indeed very similar to the suggestion in the review. I presume that the numbers reported in the manuscript (Table 1) include the warm start technique, correct? I wouldn't be too surprised if the compute time used by FermiNet would get a similar 4-8x improvement by starting from an already optimized parameters for each new configuration instance. Still, the fact that meta GNN leads to significant improvement compared to a single trained model is reassuring (thanks for pointing this out). I think it would be fantastic to add the curve to Fig. 6 that produces the values in Table 3 for N2 molecule.

---

> > > ### Author Response · Authors · 2021-11-30
> > > **Follow-up, Reviewer bjEg**
> > >
> > > Thank you for your response, we appreciate your time and that you find the improvements of the MetaGNN significant!
> > >
> > > > **The numbers reported in Table 1 include the warm start technique, correct?**
> > >
> > > The DeepErwin numbers in Table 1 are as follows:
> > > * H4+: Here we measure the training of 5000 parallel (with weight sharing) + 2000 sequential (without weight sharing) = 7000 total training steps per configuration. (plotted in Figure 3)
> > > * H4: We use the same scheme as for H4+.
> > > * H10: We measure 8192 sequential iterations per configuration. (plotted in Figure 5)
> > >
> > > None of the timings include a warm start from previously optimized parameters.
> > >
> > > We are very sorry if our original comment caused a confusion.
> > > For the camera-ready version, we will include DeepErwin with a warm start for the hydrogen chain (Figure 5). Here are the relevant numbers
> > >
> > > |H-H distance | DeepErwin (cold) | DeepErwin (warm) | PESNet | $E_\text{PESNet} - E_\text{DeepErwin(warm)}$|
> > > |---|------------------|------------------|--------|---|
> > > |1.0|-|-|-4.4258|-|
> > > |1.2|-5.1387|-5.1454|-5.1459|-0.0004|
> > > |1.4|-5.4852|-5.4870|-5.4886|-0.0016|
> > > |1.6|-5.6259|-5.6278|-5.6297|-0.0019|
> > > |1.8|-5.6618|-5.6622|-5.6632|-0.0010|
> > > |2.0|-5.6370|-5.6377|-5.6390|-0.0012|
> > > |2.4|-5.5149|-5.5164|-5.5171|-0.0007|
> > > |2.8|-5.3729|-5.3736|-5.3744|-0.0008|
> > > |3.2|-5.2466|-5.2491|-5.2499|-0.0009|
> > > |3.6|-5.1521|-5.1550|-5.1557|-0.0007|
> > >
> > > Note, the warm start model takes twice the compute time that we report in Table 1, 8192 steps to get the initialization and 8192 steps for the final optimization. Furthermore, the DeepErwin training for an interatomic distance of 1.0 diverges.
> > >
> > > > **FermiNet would get a similar 4-8x improvement by starting from an already optimized parameters.**
> > >
> > > We definitely agree that there is going to be an improvement. For the hydrogen-based system, we expect results in the range you suggested. However, similar to DeepErwin we expect the benefit to shrink with the system size due to the high-frequency wave function of heavier atoms.
> > >
> > > > **I think it would be fantastic to add the curve to Fig. 6 that produces the values in Table 3 for the N2 molecule.**
> > >
> > > Thank you for the great suggestion! We agree that adding the curve to Figure 6 will aid in communicating the benefits of the MetaGNN. We will include the change in the camera-ready version.

---

### Official Review · Reviewer_Zkwk · 2021-11-03

**Correctness:** 4
**Technical Novelty And Significance:** 3
**Empirical Novelty And Significance:** Not applicable
**Recommendation:** 8
**Confidence:** 2

**Main Review:**

Strengths
1. The proposed PESNet obtains SOTA or near-SOTA results compared to previous methods while being much faster to train.
2. Using meta learning, the PESNet is able to handle multiple geometries simultaneously.
3. The use of meta learning in this manner is novel and interesting.

Weakness
1. Recently, a number of datasets based on DFT have been created (e.g. OC20) and various GNN models have been proposed to train on such datasets (e.g. Dimenet++, Gemnet, etc). This paper, in contrast, presents an ab initio method that does not require a training dataset. While the introduction section discusses the difference in these approaches, it is unclear how the PESNet compares to the the trained methods when they are trained on very large and diverse datasets like OC20. It would be  helpful to add some discussion on this.
2. The paper uses a lot of terminology that is difficult to follow for somebody without a background in chemistry. Since ICLR is a general DL conference, I would urge the authors to change the text to make it easier to follow.


**Summary Of The Paper:**

The paper presents a new meta-learning method for solving the Schrodingers equation using neural networks. This method combines an existing neural wave function model called FermiNet together with a GNN (MetaGNN) to solve the Schrodingers equation for multiple geometries simultaneously. The MetaGNN takes the atomic graph as input and outputs a set of parameters that capture the 3D geometry of the system, which are then input to the FermiNet. The resulting method can is applicable to multiple geometries while being significantly faster to train.


**Summary Of The Review:**

This paper presents a novel ab initio method for solving the Schrodingers equation using meta-learning by combining existing neural wave function models with GNNs. The resulting method can solve the Schrodingers equation for multiple geometries while training significantly faster.

---

> ### Author Response · Authors · 2021-11-12
> **Official Response to Reviewer Zkwk**
>
> Thanks a lot for the review. We are glad that you appreciate the gained speedup and find our application of meta-learning in this context novel.
> We would like to take the opportunity to address your comments:
>
> > __..., it is unclear how the PESNet compares to the trained methods when they are trained on very large and diverse datasets like OC20.__
>
> A direct comparison between supervised and *ab-initio* methods is difficult as both target very different problems.
>
> * While the goal of supervised models is to reproduce quantum mechanical calculations in a fraction of the time it takes to perform the QM calculations, PESNet aims at performing the exact QM calculations.
> * Supervised models are inherently limited by the quality of labels which in most cases is on DFT-level accuracy. While DFT is often thought of as being precise, it has many failure cases, e.g., bond breaking. In contrast, PESNet has been shown to outperform significantly more expensive QM calculations than DFT like CCSD or CCSD(T).
> * The time to get the energy of a system also varies significantly as a single forward pass of a supervised model generally is in the order of milliseconds, evaluating the energy for a molecule *ab-initio* is in the range of hours to days.
> * As we allude to in our conclusion, an application of PESNet would be to generate labels for supervised models.
>
> To better point out the differences we added the following to related work:
> > *Note that supervised models, while architecturally related, pursue a fundamentally different goal than PESNet. Where surrogate models approximate QM calculations from data, this work focuses on performing the exact QM calculations from scratch.*
>
> ---
> >__The paper uses a lot of terminology that is difficult to follow for somebody without a background in chemistry.__
>
> We understand that a lot of the terminology might be unfamiliar to the machine learning community and tried our best to communicate to both quantum chemistry as well as machine learning audience. If there are any specific sections or paragraphs that were unclear, we would be very grateful if you could point them out to us.

---

> > ### Comment · Reviewer_Zkwk · 2021-11-22
> > **Response**
> >
> > Thank you for clarifying the differences between ab initio and supervised methods. I agree that comparing them is difficult given that their goals are different.
> >
> > For the second point, I thought that most of the paper was a little hard to follow, but I agree that it would be difficult to communicate this subject with an ML audience.

---

### Decision · Program_Chairs · 2022-01-20

**Decision:**

Accept (Spotlight)

**Comment:**

This paper builds on the success of the FermiNet neural wave function framework by pairing it with a graph neural network which predicts the parameters of neural wave function from the geometry. The resulting PESNet trains significantly faster, with no loss of accuracy. This method constitutes an important advance in ML-powered quantum mechanical calculations.

The reviewers unanimously recommend acceptance.